# Does Vigorous Physical Activity Contribute to Adolescent Life Satisfaction?

**DOI:** 10.3390/ijerph18052236

**Published:** 2021-02-24

**Authors:** František Chmelík, Karel Frömel, Dorota Groffik, Michal Šafář, Josef Mitáš

**Affiliations:** 1Faculty of Physical Culture, Palacký University Olomouc, 77111 Olomouc, Czech Republic; frantisek.chmelik@upol.cz (F.C.); karel.fromel@upol.cz (K.F.); michal.safar@upol.cz (M.Š.); 2Institute of Sport Sciences, The Jerzy Kukuczka Academy of Physical Education, 40-065 Katowice, Poland; d.groffik@awf.katowice.pl

**Keywords:** quality of life, well-being, type of physical activity, organized physical activity, IPAQ-LF

## Abstract

Background: Physical and mental health are the basis of life satisfaction (LS), even during adolescence. The aim of this study was to identify the associations between LS and types of physical activity (PA) in Czech and Polish boys and girls. Methods: The research involved 933 girls and 663 boys aged 15–19 years. LS was diagnosed using the Bern Subjective Well-Being Questionnaire and the WHO-5 Well-Being Index. The International Physical Activity Questionnaire-long form (IPAQ-LF) questionnaire was used to identify the types of weekly PA. Results: Adolescents with the highest LS had more recreational, moderate, vigorous, and total weekly PA. The strongest associations between LS and PA were in the Czech and Polish boys and girls who participated in vigorous PA (VPA). The recommendations for VPA were fulfilled by 45% of Czech and 46% of Polish boys and 40% of Czech and 50% of Polish girls, with the highest LS. The most significant positive moderator between LS and PA was participation in organized PA. Boys with the highest LS were 1.94 times more likely to meet the weekly recommendation of VPA than boys with the lowest LS. Similarly, girls with the highest LS were 1.77 times more likely to meet these recommendations. Conclusions: Promoting both current subjective well-being and organized PA, with an emphasis on achieving the VPA recommendations, may support general LS and a healthy lifestyle in adolescents.

## 1. Introduction

The associations between life satisfaction (LS), well-being, happiness, and physical activity (PA) represent a highly relevant research issue, even during adolescence [1,2,3]. The importance of the research is supported by numerous negative indicators of this developmental period, such as a significant decrease in PA and an increase in sedentary behavior with age [4,5,6,7,8], which is also associated with an increase in depression [9,10]) and other mental diseases [11,12,13]. Globally, mental health has deteriorated [14,15,16]. At the same time, the overall life satisfaction of adolescents is closely related to mental health risks [17]. “The Lancet Commission” [18] announced a call and encouraged global action to reverse the negative trend concerning the population’s mental health. The mental strain on adolescents in secondary schools is high [19,20], and it is well-known that educational strain in schools is insufficiently compensated by regeneration activities, particularly by adequate PA [21,22]. One of the most alarming findings is the repeated subjective evaluation of academic stress accompanied by increased heart rate, but without immediate compensation using physical activity [19]. Moreover, gender differences in schoolwork pressure increase with age, with 15-year-old girls reporting higher levels than boys in most countries/regions, and there has been an overall increase in schoolwork pressure [23]. Gender differences in schoolwork pressure are also likely to be reflected in LS, although Chen et al. [24] found a non-significant difference between boys and girls in their perceived LS, and descriptively, boys scored slightly higher than girls in LS.

There is no doubt that the school environment and school education significantly affect adolescents’ LS [23]. Along with a good family background, school LS is an important part of adolescents’ overall quality of life. In a study of Spanish adolescents, school connectedness, family support, academic achievement, and self-regulation were the strongest predictors of adolescents’ life satisfaction [25]. The authors also emphasized that family participation in school activities can be beneficial in adolescents’ life satisfaction. Cooperative links between the family and school seem to be crucial for adolescents’ LS [26], also in the context of supporting PA. Therefore, Kleszczewska et al. [27] believe that one of the significant predictors of LS in Polish adolescents is physical activity.

School-family cooperation in supporting adolescents’ PA may be crucial to maintain high LS among adolescents, especially during and after the pandemic. This has been shown in the Nordic countries where adolescents have high LS [28] although von Soest, Vakken, Pedersen, and Sletten [29] have pointed to a decreasing life satisfaction and subjective well-being in Norwegian adolescents during the COVID-19 pandemic.

Given the insufficient number of interventions aimed at increasing PA, and thus improving well-being and LS [30], the research on the associations between LS, well-being, and overall PA needs to be extended to include different types of PA and different PA intensities. So far, research has not focused on the associations between LS and the achievement of vigorous physical activity (VPA) recommendations. Due to the differences in the VPA of Czech and Polish adolescents and discrepancies in the educational systems [31], it is also necessary to reflect these facts in LS research.

The aim of this study was to identify the associations between LS (including general life satisfaction and current subjective well-being) and types of physical activity in Czech and Polish boys and girls. Another aim was to identify the gender differences in the achievement of PA recommendations among adolescents with different LS levels.

## 2. Materials and Methods

### 2.1. Study Design and Participants

The research was carried out in 18 secondary schools in seven regions of the Czech Republic and 22 schools in the Katowice and Wrocław regions of Poland in 2016–2017. The schools were selected on the basis of cooperation with the universities in these regions. In the selection process, the authors considered the regions, the sizes of the cities, the types of schools, and the school administrations’ interest in the research. The research used a stratified and deliberate quota sample of schools and student groups. Most frequently, two classes of students per school were involved in the research. The research sample included students whose usual school program included an Information and Communication Technologies class in the computer room. Overall, 90% to 100% of the students and their parents gave consent to the research. The research involved 663 boys and 933 girls (Table 1), and 121 participants were excluded because of a failure to complete the required data in the questionnaires. The participants were stratified by the overall LS level into four quartile-based groups (lowest, lower, higher, highest) separated for boys (lower quartile = 48.5, median = 56.2, upper quartile = 63.3) and girls (lower quartile = 43.5, median = 52.2, upper quartile = 60.2).

### 2.2. Measures

The research was organized and implemented by the same research team in the Czech Republic and Poland. The introductory information concerning the completion of the questionnaires was held in the computer room. In the beginning, all the participants registered in the “International database for research and educational support” (Indares) database (www.indares.com).

The modified Slovak version of the Bern Subjective Well-Being Questionnaire (BQ) was used to identify adolescents’ LS [32,33]. This questionnaire was used because it had been standardized for Czech and Slovak adolescents. In the study, the authors used only the results of the “General Life Satisfaction” section of the questionnaire, including the following items:My future looks good.I enjoy life more than most people.I can cope well with the realization of my life plans.Whatever happens, I can always take the best of it.I like life.My life is meaningful.My life moves in the right direction.

The answers to the items were indicated on a six-point scale from “strongly disagree” to “strongly agree” (1–6 points).

The respondents’ current subjective well-being during the previous two weeks was assessed using the WHO-5_Czech (WB) Index (https://www.psykiatri-regionh.dk/who-5/Pages/default.aspx; accessed on 22 March 2020). The Index has been used globally and is a suitable tool in research to compare well-being between groups [34]. The questionnaire included the following questions:I have felt cheerful and in good spirits.I have felt calm and relaxed.I have felt active and vigorous.I woke up feeling fresh and rested.My daily life has been filled with things that interest me.

The answers to the items were indicated on a six-point scale from “all of the time” to “at no time” (1–6 points).

For the present study, LS was determined by combining the results of both questionnaires, i.e., the overall long-term subjective expression of LS in the BQ Questionnaire and the current subjective expression of LS for the previous two weeks in the WB Index. The quartile-based level of LS was determined by the sum of the weighed points of the adjusted scoring (Figure 1).

An effort to maintain the natural habitual conditions in the schools made the authors use a subjectively integrated LS assessment approach, while respecting the economic, sociocultural, psychobiological, psychological, and functional approaches to LS [35].

The International Physical Activity Questionnaire-long form (IPAQ-LF) modified for adolescents was used to determine the participants’ weekly PA structure [36,37,38]. The Czech version of the IPAQ-LF questionnaire was developed in compliance with the applicable translation requirements and was verified in numerous research studies [39,40,41]. The coefficient of concurrent validity between the overall weekly PA (MET-min) in the IPAQ-LF questionnaire and the weekly step count (steps/week) was calculated according to Pearson’s correlation coefficient in the range of r = 0.231–0.283. The Cronbach’s alpha coefficient, measuring internal consistency and reliability of the Czech version, was α = 0.845. The data were processed following the IPAQ Scoring Manual, but all types of physical activity exceeding 180 min were recoded to this value, the MET-min for vigorous PA was multiplied by a coefficient of 6 METs, the total time of reported activity and inactivity per day was limited to 960 min/day, and the total MET-min sum per week had to be within 20,000 MET-mins. The aim of these adjustments was to not disrupt the proportional structure of weekly PA.

The weekly PA recommendation of at least three periods of no less than 20 min of VPA was determined in compliance with the general PA guidelines [42,43] and was based on previous research studies [40,41]. The VPA recommendation had to be achieved either in Part 1, “School-Related PA”, or in Part 4, “Recreation, Sport, and Leisure-Time PA”, which is a very stringent criterion, but to a certain extent it eliminates the known overestimation of time and frequency of PA types in the IPAQ-LF questionnaire [44]. The participants completed the questionnaires in the following order: 1. Bern Questionnaire, 2. IPAQ-LF Questionnaire, and 3. WB 5 Index.

### 2.3. Data Analysis

The statistical analyses were performed in Statistica, version 13 (StatSoft, Prague, Czech Republic) and SPSS, version 25 (IBM Corp., Armonk, NY). First, the basic descriptive statistics were computed. Next, the Kruskal–Wallis test was applied to analyze the differences in the aggregate PA of each group of participants and the contingency tables to assess the differences in the achievement of PA recommendations. The binary logistic regression with the standard enter method (where all independent variables were entered into the equation simultaneously) was used to assess the odds of achieving VPA recommendations. The *η*^2^ and *d* effect size coefficients were evaluated as follows: 0.01 ≤ *η*^2^ < 0.06—small effect size, 0.06 ≤ *η*^2^ < 0.14—medium effect size, *η*^2^ ≥ 0.14—large effect size, 0.2 ≤ *d* < 0.5—small effect size, 0.5 ≤ *d* < 0.8—medium effect size, and *d* ≥ 0.8—large effect size. Based on previous experience, the threshold value for assessing differences as logically significant was set at >2000 MET-min and at >10% for the achievement of weekly PA recommendations. 

### 2.4. Ethical Principles

The present study was approved by the Ethics Committee of the Faculty of Physical Culture, Palacký University Olomouc under reference number 24/2012. The involvement of all schools and participants in the Czech Republic and Poland was voluntary. The participants were included in the study on the basis of informed consent, including parental consent. All data in this study were collected anonymously.

## 3. Results

### 3.1. Initial Characteristics of LS and PA Among Czech and Polish Adolescents

Czech boys reported the highest LS (sum of points for both questionnaires) at Mdn ± IQR = 56.20 ± 13.44 (Polish boys 55.92 ± 16.00), whereas Polish girls reported the lowest LS at Mdn ± IQR = 49.52 ± 18.80 (Czech girls 54.04 ± 14.72). Statistically significant differences (H = 55.77, *p* < 0.001, *d* = 0.37) were observed between Polish and Czech girls (*p* < 0.001) in favor of Czech girls and between Polish girls and Polish boys (*p* < 0.001) in favor of Polish boys.

Polish boys reported the highest weekly PA (according to the MET-min) at Mdn ± IQR = 5548 ± 5931 (Czech boys 4453 ± 5348), whereas Czech girls reported the lowest weekly PA at Mdn ± IQR = 3971 ± 4521 (Polish girls 4260 ± 6230). In the overall weekly PA, no statistically significant differences were observed between Polish and Czech girls or between Polish and Czech boys. However, Polish girls reported more school PA (*p* = 0.006), transport PA (*p* = 0.040), recreational PA (*p* = 0.003), and moderate PA (*p* = 0.041) than Czech girls.

The differences in LS and PA by country and gender were also confirmed in the associations between total weekly PA (MET-min) and LS (sum points) (Figure 2).

### 3.2. Associations between Life Satisfaction and Types of Weekly Physical Activity

Generally, adolescents in both countries with the highest LS reported more statistically significant recreational PA (H(3,1596) = 49.51; *p* < 0.001; *η*^2^ = 0.029), vigorous PA (H(3,1596) = 42.05; *p* < 0.001; *η*^2^ = 0.025), moderate PA (H(3,1596) = 17.92; *p* < 0.001; *η*^2^ = 0.009), and overall PA (H(3,1596) = 26.08; *p* < 0.001; *η*^2^ = 0.014) than adolescents with the lowest LS. In the case of transport PA (H(3,1596) = 1.61; *p* = 0.658; *η*^2^ = 0.001) and walking (H(3,1596) = 3.23; *p* = 0.358; *η*^2^ = 0.001) the associations were not significant.

As far as gender and country differences are concerned, identical statistically significant associations between LS and PA were observed among Czech and Polish girls, specifically in recreational PA, VPA, moderate PA, and total PA (Table 2). In the group of boys, the associations were not so clear. Among Czech boys, statistically significant associations were only observed in recreational PA, while they were only observed in VPA among Polish boys (Table 2).

The strongest associations between LS and PA types were observed in VPA. These associations (between groups with the lowest and highest LS) were also statistically significant between LS and the achievement of the recommendation of at least three periods of no less than 20 min of VPA in Czech girls (χ^2^ = 11.36; *p* = 0.009; *d* = 0.311), Polish girls (χ^2^ = 11.06; *p* = 0.011; *d* = 0.311), and Polish boys (χ^2^ = 19.85; *p* < 0.001; *d* = 0.471), but not in Czech boys (χ^2^ = 4.85; *p* = 0.183; *d* = 0.232). The difference of 13.7 percentage points between boys with the lowest and highest LS was considered as logically significant (Figure 3).

Statistically significant differences between the lowest and highest LS quartiles of participants involved in organized PA were observed in Czech girls (χ^2^ = 17.69; *p* < 0.001; *d* = 0.615; difference between the lowest and highest LS groups *p* < 0.001), Polish girls (χ^2^ = 10.78; *p* = 0.013; *d* = 0.407; intergroup difference *p* = 0.002), and Polish boys (χ^2^ = 7.97; *p* = 0.047; *d* = 0.406; intergroup difference *p* = 0.008) (Figure 4). The difference between the highest and lowest LS quartiles among Czech boys (χ^2^ = 8.60; *p* = 0.035; *d* = 0.532; intergroup difference *p* = 0.055) with active participation in organized PA can be considered logically significant concerning the difference of 15.7 p.p.

Boys with the highest LS had 1.94 times higher chances of meeting the recommendation of at least three periods of no less than 20 min of VPA per week than boys with the lowest LS (OR = 1.94; CI 1.21–3.11; *p* = 0.006; *R*^2^ = 0.026). Similarly, girls with the highest LS had 1.77 times higher chances (OR = 1.77; CI 1.14–2.74; *p* = 0.011; *R*^2^ = 0.012) of meeting the VPA recommendation. The control variables (age, BMI, country, time stages of research, socioeconomic status, and participation in organized PA) involved in the model did not show a significant effect on the achievement of the VPA recommendation among boys and girls with the highest LS and did not decrease the predictive significance of LS.

## 4. Discussion

The most serious finding is that Czech and Polish girls with the highest LS reported more recreational PA, moderate PA, VPA, and total PA than girls with the lowest LS. Among boys, the results were not so noticeable, although boys with the highest LS showed associations with higher recreational PA, VPA, and total PA overall for both countries. The association between the highest LS and VPA in both boys and girls requires considerable attention. However, studies addressing the associations between LS and VPA among adolescents are rare. The long-term contribution of VPA on higher LS, when compared with moderate PA, was confirmed by Kaczmarek et al. [45], but only in students aged 18 to 45 years. Lithuanian authors Slapšinskaitė, Lukoševičiūtė, and Šmigelskas [2] considered VPA in boys to be a predictor of better life satisfaction, whereas in girls, moderate-to-vigorous PA (MVPA) was considered to be a risk factor for lower LS.

The associations observed between the highest LS and VPA in girls need to be interpreted with caution, mainly because the current situation and trends in VPA are unfavorable, especially in girls. Moreover, the differences in VPA between boys and girls increase with age, often to girls’ detriment [23]. According to the European Commission [46], 58% of respondents in Europe had not performed any vigorous PA in the previous week, and there were also considerable gender differences. Between the ages of 15 and 24, no VPA was performed by 31% of boys and 53% of girls. No VPA was performed by 58% of adolescents in the Czech Republic and by 60% of adolescents in Poland. Since 2013, there has been an increase of 4 p.p. in those who do not perform any VPA. A decrease of 3 p.p. has been observed among those who perform VPA. In this context, a very serious finding is that health-related LS decreases with age, especially in girls [47], and also in Europe [48].

The strong associations identified between LS and VPA are likely to be significantly affected by active participation in organized PA. In the present study, only 19.9% of boys who participated in organized PA showed no VPA during the week, compared to 32.7% of boys who did not participate in organized PA. Similarly, only 26.3% of actively participating girls reported no VPA during the week, compared to 41.1% of girls who did not participate in organized PA.

Boys’ participation in organized sports, including VPA, was emphasized by Lagestad et al. [49], mainly to prevent boys from leaving organized sport during adolescence. This is consistent with the results of Grao-Cruces et al. [50], who observed that boys with lower LS were 2.7 times more likely to have a low intention to be physically active. However, the danger of quitting active participation in organized PA is even greater among adolescent girls. They show a lower health-related quality of life and LS than boys. Therefore, this group’s crucial aspects include PA and the consumption of healthy food [51]. In terms of the contribution of active participation in organized PA to adolescents’ LS, it may also be important to support self-organized activities in addition to structured activities, as recommended, for example, by Wiium and Säfvenbom [52].

The fact that girls reported lower life satisfaction corresponds to similar findings by other authors [53,54]. Adolescent girls are at the greatest risk, especially in the context of experiencing negative feelings and low self-concept and life satisfaction, and in relation to self-harming [55]. In the Nordic countries, which have an overall high level of LS [56], the group at the greatest risk of decreasing LS is 15-year-old girls [28]. It turns out that age and gender have a high level of discriminatory power for life satisfaction and self-rated health and are both significant moderators between adolescents’ health and well-being [57]. Efforts to improve physical fitness, which is also dependent on VPA, should also be an important strategy in promoting health-related quality of life in younger adolescents aged 8–11 years, as emphasized by Morales et al. [58]. 

Another important finding relates to the associations between LS and the achievement of PA recommendations. Regarding the analysis of the associations between LS and types of PA, the most relevant recommendation is that of at least three periods of no less than 20 min of VPA per week. The most preferred general recommendation is ≥60 min/day of MVPA [42,43]. Most adolescents do not meet these recommendations. Currie et al. [59] concluded that the daily recommendation of ≥60 min/day of MVPA is not achieved by 77–85% of European adolescents. A decline was also observed in the following years, as only 19% of adolescents achieved the recommended 60 min of MVPA daily. Moreover, a lower level of MVPA and VPA was observed in adolescents from poorer families in most countries [23]. The fact that adolescents actively participating in organized PA are more likely to achieve the PA recommendations has been confirmed in Central European and Northern European countries [60,61].

The degree to which the achievement of PA recommendations and well-being increase life satisfaction, and the degree to which life satisfaction causes better well-being and a better attitude towards physical activity, is greatly individual and multidimensional. It is not easy to quantify PA’s benefits associated with an increased quality of life and LS [62]. Interventions aimed at the associations between PA, well-being, and LS, and comparative studies on LS among adolescents between countries, do not always bring clear evidence [1,63]. This evidence is found primarily in partial research studies. For example, according to the evidence concerning the positive association between PA and happiness, even only 10 min of PA or exercise per week makes a difference in the level of happiness [3]. Therefore, research on the associations between LS and the overall level of PA needs to include different types, frequencies, volumes, and intensities of PA.

The importance of research aimed at the associations between symptoms of depression, LS, and various PA types in adolescents in terms of volume and intensity of PA is highlighted by Kleppang et al. [64]. Especially during and after the pandemic, new facts about the associations between the level of LS and types of PA may help to eliminate some of the consequences of the pandemic on adolescents’ lifestyle.

Future research studies should identify when the associations between adolescents’ LS and VPA are weakened or not present at all. It would also be fascinating to investigate these associations in adolescents on the basis of the following quartet: vigorous PA–physical fitness–mental condition–life satisfaction.

### Strengths and Limitations

One of the study’s strengths is the analysis of the associations between LS and PA among adolescents within educational systems in the Czech Republic and Poland. Another strength is respecting the different types of PA and PA recommendations concerning adolescents’ LS. A main methodological limitation is the deliberate sampling method and subjective estimates of weekly PA, well-being, and LS.

## 5. Conclusions

The positive associations between VPA and LS in adolescent boys and girls emphasize the importance of VPA in weekly school-based PA, out-of-school PA, and organized PA. Boys and girls from both countries with the highest LS also participated the most in organized PA. Associations between LS and achievement of the VPA recommendation were statistically significant in Czech and Polish girls and Polish boys. School-based PA programs should support the achievement of at least three periods of no less than 20 min of VPA per week as part of the daily recommendation of 60 min of PA.

Supporting current subjective well-being in schools and organized PA in and after school, with an emphasis on achieving the VPA recommendations, may support general LS and a healthy lifestyle in adolescents.

## Figures and Tables

**Figure 1 ijerph-18-02236-f001:**
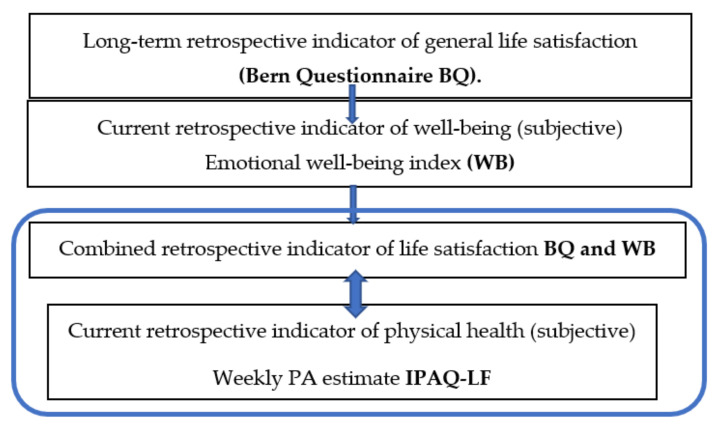
Research Design.

**Figure 2 ijerph-18-02236-f002:**
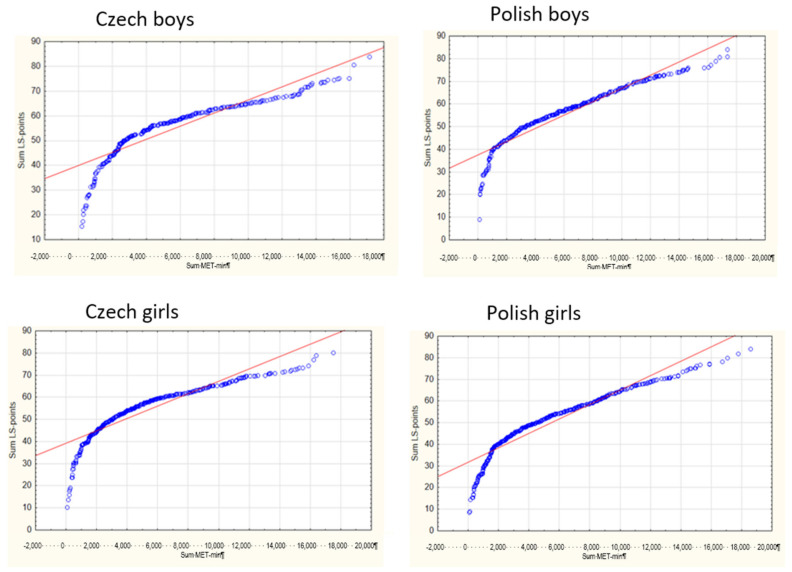
Association between life satisfaction (points) and total weekly physical activity (PA) (MET-min) among Czech and Polish boys and girls.

**Figure 3 ijerph-18-02236-f003:**
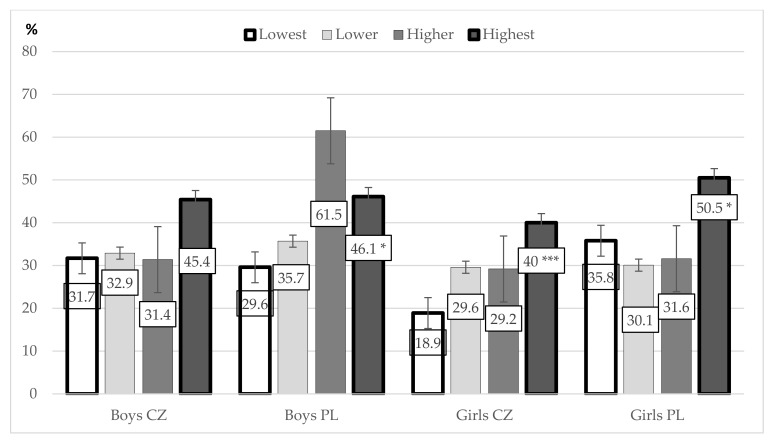
Associations between life satisfaction quartiles (lowest, lower, higher, highest) and the achievement (percentage of those achieving) of the recommendation for vigorous physical activity (VPA) (at least three periods of no less than 20 min of VPA per week) among Czech and Polish boys and girls. Significant difference between the groups with the lowest and highest LS: * *p* < 0.05; *** *p* < 0.001.

**Figure 4 ijerph-18-02236-f004:**
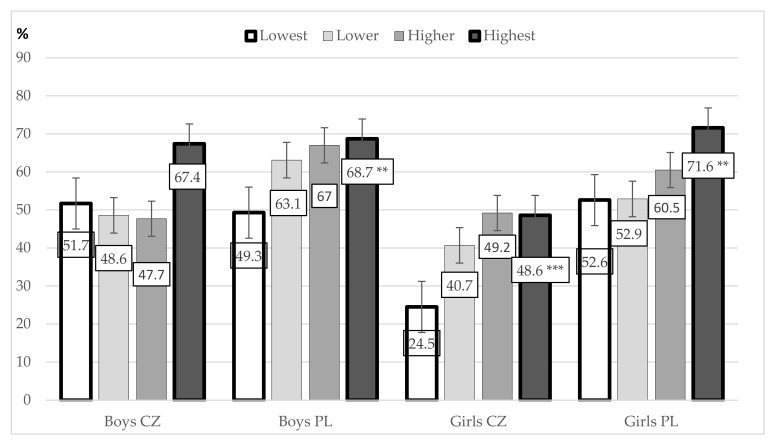
Associations between life satisfaction quartiles (lowest, lower, higher, highest) and active participation in organized PA (percentage of those participating) among Czech and Polish boys and girls. Significant difference between the groups with the lowest and highest life satisfaction (LS): ** *p* < 0.01; *** *p* < 0.001.

**Table 1 ijerph-18-02236-t001:** Sample Characteristics.

Characteristics	Country	n	Age (Years)		Weight (kg)		Height (cm)		BMI (kg·m^−2^)	
			M	SD	M	SD	M	SD	M	SD
Boys	CZ	302	16.72	1.08	69.80	12.64	178.90	7.73	21.74	3.24
	PL	361	16.31	0.74	68.98	13.88	177.26	7.53	21.88	3.73
Girls	CZ	466	16.61	1.01	59.24	9.63	167.61	6.47	21.06	3.01
	PL	467	16.30	0.65	56.88	8.45	165.73	5.98	20.70	2.83

M—mean; SD—standard deviation; BMI—Body Mass Index; CZ—Czech Republic; PL—Poland.

**Table 2 ijerph-18-02236-t002:** Associations between life satisfaction (lowest, lower, higher, and highest) and types of weekly physical activity (MET-min) among Czech and Polish boys (CZ *n* = 302, PL *n* = 361) and girls (CZ *n* = 466, PL *n* = 467).

Type of Physical Activity	Gender	Country	Life Satisfaction								*H*	*p*	*η* ^2^
			Lowest		Lower		Higher		Highest				
			*Mdn*	*IQR*	*Mdn*	*IQR*	*Mdn*	*IQR*	*Mdn*	*IQR*			
Transportation	Boys	CZ	672	1430	836	1665	743	1062	695	1436	0.66	0.884	0.008
		PL	990	1905	875	1621	594	1980	945	1370	2.72	0.437	0.001
	Girls	CZ	767	1155	990	1451	693	1478	924	1400	2.26	0.520	0.002
		PL	660	1215	660	1370	718	1908	792	1919	1.19	0.773	0.004
Recreation	Boys	CZ	1049	2592	1128	1602	986	1899	1833	2198	14.62 ^a^	0.002	0.039 *
		PL	693	1760	846	2192	931	2435	1188	2478	4.06	0.255	0.003
	Girls	CZ	587	1337	990	1871	1188	1722	1374	1881	19.11 ^a^	<0.001	0.035 *
		PL	491	1551	630	1485	692	1581	938	2226	14.06 ^a^	0.003	0.024 *
Vigorous	Boys	CZ	900	2160	840	2040	765	1980	1500	2760	7.57	0.056	0.015 *
		PL	450	2250	720	2220	1980	2820	1440	3180	17.29 ^a^	<0.001	0.040 *
	Girls	CZ	180	1170	540	1620	615	1440	1020	1860	13.52 ^a^	0.004	0.023 *
		PL	360	1800	420	1800	570	1680	1260	2820	12.60 ^a^	0.006	0.021 *
Moderate	Boys	CZ	1160	2530	1210	2640	1239	1700	2130	3120	6.45	0.092	0.012 *
		PL	1300	3220	1550	2528	1740	3310	1460	2700	2.14	0.544	0.002
	Girls	CZ	730	1020	1015	1660	890	1680	1270	1695	11.52 ^a^	0.009	0.018 *
		PL	1005	1860	1260	2200	1290	2140	1720	2460	10.04 ^a^	0.018	0.015 *
Walking	Boys	CZ	1254	2285	1436	2063	1295	2673	1254	1980	0.87	0.832	0.007
		PL	1617	2987	1378	2384	990	2574	1683	2409	3.41	0.332	0.001
	Girls	CZ	1411	2492	1964	2921	1246	2541	1749	2541	5.97	0.113	0.006
		PL	1551	2541	1221	2360	1452	2805	1617	2739	0.67	0.880	0.005
Total (vigorous, moderate, walking)	Boys	CZ	4106	5629	4241	4590	4017	4368	5552	6432	6.76	0.080	0.013 *
		PL	4581	5877	5122	5540	6630	5774	5623	6049	4.72	0.193	0.005
	Girls	CZ	3173	3716	4421	5061	3890	3887	5063	5234	12.98 ^a^	0.005	0.022 *
		PL	3720	5200	4546	6287	4541	5296	4782	6493	8.27 ^a^	0.041	0.011 *

*Mdn*—median values; *IQR*—interquartile ranges; *H*—Kruskal–Wallis test; ^a^ significant difference between groups (lowest and highest life satisfaction); *η*^2^—effect size; *p*—significance level; * small effect size.

## Data Availability

The data presented in this study are available on request from the corresponding author.

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
