# Peer review of "Does Vigorous Physical Activity Contribute to Adolescent Life Satisfaction?"

_ijerph, 2021, doi:10.3390/ijerph18052236_

Round 1
Reviewer 1 Report
This is an paper with interesting data in this field of Life Satisfaction and Physical Activity. But what I miss is a more dimensional visual view. I would challenge the authors to extend their analysis with graphics that show the 2D or correlations between LS and PA with a variable like BMI because I assume that higher BMI will show lower PA and LS scores and their might be also a gender or cultural difference . If you would be able to show in color the gender effect in this extra task, I think the reader would be much more interested in this topic.
Author Response
Comments and Suggestions for Authors
This is an paper with interesting data in this field of Life Satisfaction and Physical Activity. But what I miss is a more dimensional visual view. I would challenge the authors to extend their analysis with graphics that show the 2D or correlations between LS and PA with a variable like BMI because I assume that higher BMI will show lower PA and LS scores and their might be also a gender or cultural difference . If you would be able to show in color the gender effect in this extra task, I think the reader would be much more interested in this topic.
Response:
Dear reviewer, we would like to thank you for your feedback and valuable recommendations. We extended the analyses and added a figure showing the associations between LS and PA reflecting country and gender of the participants.
Due to the low number of students, we decided not to extend the results to analyzes groups with different BMI. The logistic regression at the end of the Results chapter suggests an insignificant effect of BMI on the association between PA and LS. Similar results arise from partial analyzes of overweight or obese participants.
We tried to maintain the most natural educational environment, so it was not possible to objectively measure the height and weight of participants in student teaching practice. These were accurately measured only as part of the weekly PA monitoring. In addition, some school leaders fundamentally reject any somatometric measurements.
Reviewer 2 Report
Dear Authors,
Congratulations on your research and thank you for your patience regarding this revision.
I will make my considerations by sections.
Introduction
When reading the first part of the Introduction, the first glimpse made me think the paper was about physical activity and mental diseases (L. 28-44). I even underlined some of them to catch up, and those I have highlighted were: depression; mental diseases; mental health; mental strain; academic stress; schoolwork pressure. Then, in the second part (L. 45-68), the key-word “family” drew my attention.
After reading the title and the abstract, I was expecting some background info about Physical activity (all types and intensity) and Life Satisfaction. Please do not get me wrong. Despite being well written, the Introduction does not provide adequate context for their topic, even though I find the topic important and timely where other relevant researches are cited.
The research question poses some doubts to me as they aim to identify associations between LS and Current subjective Well-being, and only LS is presented in the results. I can understand that LS was drawn by the results of different questionnaires (BQ and WB), but only the LS has its results presented and discussed. Also, the authors aim to identify gender differences regarding LS levels. In the Introduction, the authors only refer to gender in one paragraph (L. 43). This objective is not justified by any gap in previous research nor presented in the Introduction, thus not clearly stating the purpose and the hypothesis of the manuscript.
This might pose some difficulty for the general readership of the journal to find the topic meaningful.
Regarding the references, authors use original and up to date references, which are all adequately used and cited. Nevertheless, despite not considering they were abusively used to meet self-citation purposes, authors have 11 papers in references where they have participated (17%).
Please check the author's commentary for further information.
Methods
The study design is adequate. Nonetheless, in the methods section, authors refer to “The research used a stratified and deliberate quota sample of schools and student groups” (L. 75). It would be welcome to know the deliberate quota and the procedure to calculate it. We can infer that each of the 40 schools participated with ±40 students (despite the imbalance between girls and boys).
This was the first time we knew that the participants would be treated as independent samples (the two countries). Nevertheless, no information on the Introduction or objectives refers to this detail. Getting ahead on my criticism, I do not fully understand why they do not act as the same participant population with boys and girls, same age, LS levels, different types of practice, but the same tendency of growing LS with more VPA. Authors make some country comparisons, which were not introduced, nor discussed. If there is any advantage in having both countries separated, please give a more detailed insight into it.
The authors should provide information on the points of adjusted scoring for the LS quartile values. Also, each quartile's cut-off values would be welcome to get an insight into LS's Raw data (got by the sum of the points of adjusted score). I also would like to ask if the items were weighed, as BQ contributes with more total points than WB.
Besides, I have questioned myself whether I would have used a Follow-up analysis of the Kruskal-Wallis test (the same as the pairwise comparisons in an ANOVA test). The author found differences in the groups defined by the LS quartiles, but we cannot get the overall picture; which of them is different from? (Lowest, lower, higher, highest).
I think that d is not the best measure of effect size for a Kruskal-Wallis comparison, moreover if a follow-up analysis is made.
Results
Even though the authors have the objective of identifying the associations between general life satisfaction, current subjective well-being, and PA, L. 117 indicates that LS is the combination of the results of 2 questionnaires. In L. 188 describes the BQ results in a manner that confuses the objectives of the manuscript. What does this information add besides describing all those percentages? Readers cannot find the criteria for that presentation. Moreover, comparing the two countries is not in the objectives. The authors then refer that “According to the WB Index, more than half of adolescents showed depressive symptoms (56.6% of Czech boys, 53.5% of Polish boys, and 52.6% of Czech girls”. This info should have more attention from the authors as the Mdn values of life satisfaction were ≈50 or more (assuming that this value falls on the Higher/highest quartile). That is why LS's best total score should be presented because the division into quartiles is made among the data you enter. Remember the GIGO. So, if all respondents are depressive, you will also get Higher/highest quartiles to work with.
Please make more clear what the authors intend when comparing both countries. In L. 202, “In the overall weekly PA, no statistically significant differences were observed between Polish and Czech girls or between Polish and Czech boys. However, Polish girls reported more school PA (p = 0.006), transport PA (p = 0.040), recreation PA (p = 0.003), and moderate PA (p = 0.041) than Czech girls”.
Data presentation gets interesting in 3.2. At this moment, readers can state the clear objective of the present work. A graphic here would be welcome. As one variable is growing, the other variables too. A two-axis graphic would give a clearer view of the data.
Again in L. 214, the two-country comparison is made.
Table 2 is hard to read and to draw a clear framework of the data tendency. I have addressed my concerns before regarding these comparisons. My best guess is that author should run pairwise comparisons between the lowest to highest groups. Then run the statistics and present the data. Imagine if the differences are in the lowest to lower. Even with a non-parametric test like KW, multiple comparisons can be computed. The conclusions driven must be different if the differences arise from the bottom quarter to the top quarter. I hope you understand what I intend. Also, insert the unit of the data presented. I guess that is the MET.
If it helps, why don’t you merge the lower quartiles and the higher quartiles in one? If you are not presenting where the differences are coming from, the final data must be the same.
When presenting the odds-ratios L.252, authors report the variables that were not significant predictors, which is as important as knowing the significant predictors. One of those was Organized PA. If organized PA does not improve the model of which the O-R are calculated, why is it used in the comparisons between the lowest and highest LS (L.240)? “Statistically significant differences between the lowest and highest LS of participants involved in organized PA were observed in Czech girls, Polish girls and Polish boys”
Authors present the O-R and its confidence interval and might consider reporting the beta values and their standard errors and general statistics about the model (such as goodness-of-fit statistics).
Comment
Authors report results of Czech and Polish (and boys) girls with the same tendency. Once more, why not have them as a part of the same group of participants.
The discussion commences with the main topic, LS and PA; however, in L. 283-288 some data is discussed, which was not presented in results, or I could not find it.
I have drawn some attention to several points that could be improved in this manuscript with very interesting data. The discussion focuses only on LS and PA, which are the main topics of this manuscript and those who will gather readers' attention. All besides that brings some confusion and makes the manuscript hard to read. So, as I referred before, all issues that are not concurring with LS and PA could be discarded (e.g. Comparison between the two countries).
Please take a look at the objectives as subjective well-being is not clearly presented in the result, nor is it discussed. You might want better to explore the gender differences, mostly in the results.
Authors identify the study’s limitations and weaknesses
Conclusion
The conclusions are succinct, but not all research questions were answered, for instance “identify the gender differences in the achievement of PA recommendations among adolescents with different LS levels”
Figures and Tables
The information in the tables and figures is not easy to interpret. I call my attention to this point.
Abstract
Check the observations and then reorganize the abstract
References
Authors use original and up to date references, which are all adequately used and cited.
Author Response
Comments and Suggestions for Authors
Dear Authors,
Congratulations on your research and thank you for your patience regarding this revision.
I will make my considerations by sections.
Response:
Dear reviewer, we would like to thank you for your feedback and valuable recommendations. We have reflected them throughout the manuscript, and we believe that we improved the quality and the clarity of the paper thanks to your contribution.
Introduction
When reading the first part of the Introduction, the first glimpse made me think the paper was about physical activity and mental diseases (L. 28-44). I even underlined some of them to catch up, and those I have highlighted were: depression; mental diseases; mental health; mental strain; academic stress; schoolwork pressure. Then, in the second part (L. 45-68), the key-word “family” drew my attention.
Response:
Thank you for your comments. We have added some information in order to provide better context for the topic. We added the text “At the same time, the overall life satisfaction of adolescents is closely related to mental health risks [17]” and we excluded the text “One of the most alarming findings is the repeated subjective evaluation of academic stress accompanied by increased heart rate without immediate compensation using physical activity [18]“. Unfortunately, we lack studies that adequately address the associations between life satisfaction and PA types. Therefore we left more general topics related to overall health indicators in adolescents.
After reading the title and the abstract, I was expecting some background info about Physical activity (all types and intensity) and Life Satisfaction. Please do not get me wrong. Despite being well written, the Introduction does not provide adequate context for their topic, even though I find the topic important and timely where other relevant researches are cited.
The research question poses some doubts to me as they aim to identify associations between LS and Current subjective Well-being, and only LS is presented in the results. I can understand that LS was drawn by the results of different questionnaires (BQ and WB), but only the LS has its results presented and discussed. Also, the authors aim to identify gender differences regarding LS levels. In the Introduction, the authors only refer to gender in one paragraph (L. 43). This objective is not justified by any gap in previous research nor presented in the Introduction, thus not clearly stating the purpose and the hypothesis of the manuscript.
This might pose some difficulty for the general readership of the journal to find the topic meaningful.
Response:
Thank you for the comment. We added “Gender differences in schoolwork pressure are also likely to be reflected although Chen et al. [24] found a non-significant difference between boys and girls in their perceived LS, and descriptively, boys scored slightly higher than girls in LS”. We also revised the aim in order to clarify it.
Regarding the references, authors use original and up to date references, which are all adequately used and cited. Nevertheless, despite not considering they were abusively used to meet self-citation purposes, authors have 11 papers in references where they have participated (17%).
Response:
We understand that the number of citations of our previous works is high. We were struggling to find other relevant references. To decrease the self-citation rate, we decided to withdraw the references of Žatka et al. (2018), Kudlacek et al. (2020), and Vašíčková et al. (2013).
Please check the author's commentary for further information.
Methods
The study design is adequate. Nonetheless, in the methods section, authors refer to “The research used a stratified and deliberate quota sample of schools and student groups” (L. 75). It would be welcome to know the deliberate quota and the procedure to calculate it. We can infer that each of the 40 schools participated with ±40 students (despite the imbalance between girls and boys).
Response:
We didn’t perform any calculation. As described, we considered the regions, size of the cities, type of schools, and school administration interest. If our formulation is misleading, it is possible to remove the sentence “The research used a stratified and deliberate quota sample of schools and student groups”. We added the text: “Most frequently, two classes of students per school were involved in the research”.
This was the first time we knew that the participants would be treated as independent samples (the two countries). Nevertheless, no information on the Introduction or objectives refers to this detail. Getting ahead on my criticism, I do not fully understand why they do not act as the same participant population with boys and girls, same age, LS levels, different types of practice, but the same tendency of growing LS with more VPA. Authors make some country comparisons, which were not introduced, nor discussed. If there is any advantage in having both countries separated, please give a more detailed insight into it.
Response:
We tried to make this more clear by adding the following text into the introduction: “Due to the differences in VPA of Czech and Polish adolescents and discrepancies in the educational systems [31], it is also necessary to reflect these facts in LS research.”
The authors should provide information on the points of adjusted scoring for the LS quartile values. Also, each quartile's cut-off values would be welcome to get an insight into LS's Raw data (got by the sum of the points of adjusted score). I also would like to ask if the items were weighed, as BQ contributes with more total points than WB.
Response:
Thank you. We improved the description of the quartiles as we added lower, median and upper quartile cut-off values. The items were weighed – we added the information in the text.
Besides, I have questioned myself whether I would have used a Follow-up analysis of the Kruskal-Wallis test (the same as the pairwise comparisons in an ANOVA test). The author found differences in the groups defined by the LS quartiles, but we cannot get the overall picture; which of them is different from? (Lowest, lower, higher, highest).
I think that d is not the best measure of effect size for a Kruskal-Wallis comparison, moreover if a follow-up analysis is made.
Response:
Thank you for the warning, the mention in the text was insufficient. We have supplemented an index to clearly distinguish significant differences between the lowest and highest quartiles of LS in the table and we also amended the corresponding text. We agree that “d” should be substituted, and we adapted the effect size for Kruskal-Wallis comparison to eta squared.
Results
Even though the authors have the objective of identifying the associations between general life satisfaction, current subjective well-being, and PA, L. 117 indicates that LS is the combination of the results of 2 questionnaires. In L. 188 describes the BQ results in a manner that confuses the objectives of the manuscript. What does this information add besides describing all those percentages? Readers cannot find the criteria for that presentation. Moreover, comparing the two countries is not in the objectives. The authors then refer that “According to the WB Index, more than half of adolescents showed depressive symptoms (56.6% of Czech boys, 53.5% of Polish boys, and 52.6% of Czech girls”. This info should have more attention from the authors as the Mdn values of life satisfaction were ≈50 or more (assuming that this value falls on the Higher/highest quartile). That is why LS's best total score should be presented because the division into quartiles is made among the data you enter. Remember the GIGO. So, if all respondents are depressive, you will also get Higher/highest quartiles to work with.
Response:
We agree. For simplicity, we have excluded the paragraph.
Please make more clear what the authors intend when comparing both countries. In L. 202, “In the overall weekly PA, no statistically significant differences were observed between Polish and Czech girls or between Polish and Czech boys. However, Polish girls reported more school PA (p = 0.006), transport PA (p = 0.040), recreation PA (p = 0.003), and moderate PA (p = 0.041) than Czech girls”.
Response:
We added an explanation into the introduction. “Due to the differences in VPA of Czech and Polish adolescents and discrepancies in the educational systems [31], it is also necessary to reflect these facts in LS research.” These are characteristics of weekly physical activity, which should be considered when interpreting the results.
Data presentation gets interesting in 3.2. At this moment, readers can state the clear objective of the present work. A graphic here would be welcome. As one variable is growing, the other variables too. A two-axis graphic would give a clearer view of the data.
Response:
Thank you for this note, we adapted the introduction, aim, and results to clarify these discrepancies and we also included a new graphic to present the requested results of analyses.
Again in L. 214, the two-country comparison is made.
Table 2 is hard to read and to draw a clear framework of the data tendency. I have addressed my concerns before regarding these comparisons. My best guess is that author should run pairwise comparisons between the lowest to highest groups. Then run the statistics and present the data. Imagine if the differences are in the lowest to lower. Even with a non-parametric test like KW, multiple comparisons can be computed. The conclusions driven must be different if the differences arise from the bottom quarter to the top quarter. I hope you understand what I intend. Also, insert the unit of the data presented. I guess that is the MET.
Response:
Thank you. We have modified the Table 2 to make it self-explanatory. We hope it makes the results clear now. Also, the unit MET-min was added to the title of the Table 2.
If it helps, why don’t you merge the lower quartiles and the higher quartiles in one? If you are not presenting where the differences are coming from, the final data must be the same.
Response:
We have highlighted the comparison of the groups Q1 and Q4 by an index. We believe that this division is more telling than the division based on median value. Differences are indicated now.
When presenting the odds-ratios L.252, authors report the variables that were not significant predictors, which is as important as knowing the significant predictors. One of those was Organized PA. If organized PA does not improve the model of which the O-R are calculated, why is it used in the comparisons between the lowest and highest LS (L.240)? “Statistically significant differences between the lowest and highest LS of participants involved in organized PA were observed in Czech girls, Polish girls and Polish boys”
Authors present the O-R and its confidence interval and might consider reporting the beta values and their standard errors and general statistics about the model (such as goodness-of-fit statistics).
Response:
Due to the number of results, we no longer wanted to present further details (tables or odds ratio plots) on the results of logistic regression. Unfortunately, a deeper analysis of this serious issue would require a separate article.
Response:
Comment
Authors report results of Czech and Polish (and boys) girls with the same tendency. Once more, why not have them as a part of the same group of participants.
The discussion commences with the main topic, LS and PA; however, in L. 283-288 some data is discussed, which was not presented in results, or I could not find it.
I have drawn some attention to several points that could be improved in this manuscript with very interesting data. The discussion focuses only on LS and PA, which are the main topics of this manuscript and those who will gather readers' attention. All besides that brings some confusion and makes the manuscript hard to read. So, as I referred before, all issues that are not concurring with LS and PA could be discarded (e.g. Comparison between the two countries).
Please take a look at the objectives as subjective well-being is not clearly presented in the result, nor is it discussed. You might want better to explore the gender differences, mostly in the results.
Response:
Thank you for your very serious suggestions. We also made partial adjustments and additions in the abstract. Combining boys and girls from both countries would be clearer and easier for readers. However, not using the difficult comparison between countries would be an "escape" from a more difficult but important solution to the problem. We decided to keep the analysis separated for the two countries.
Authors identify the study’s limitations and weaknesses
Conclusion
The conclusions are succinct, but not all research questions were answered, for instance “identify the gender differences in the achievement of PA recommendations among adolescents with different LS levels”
Response:
We added more results of the analyses to the conclusions.
Figures and Tables
The information in the tables and figures is not easy to interpret. I call my attention to this point.
Response:
We rearranged the tables and figures to make them more reader friendly.
Abstract
Check the observations and then reorganize the abstract
Response:
We partially adjusted the abstract.
References
Authors use original and up to date references, which are all adequately used and cited.
Round 2
Reviewer 1 Report
i think you reflected satisfactory on my comments. I hope you will use my 2 D visual suggestions for your next project